# Calcium Binds to Transthyretin with Low Affinity

**DOI:** 10.3390/biom12081066

**Published:** 2022-08-02

**Authors:** Cristina Cantarutti, Maria Chiara Mimmi, Guglielmo Verona, Walter Mandaliti, Graham W. Taylor, P. Patrizia Mangione, Sofia Giorgetti, Vittorio Bellotti, Alessandra Corazza

**Affiliations:** 1Department of Medicine, University of Udine, 33100 Udine, Italy; cristina.cantarutti@uniud.it (C.C.); walter.mandaliti@uniud.it (W.M.); 2Istituto Nazionale Biostrutture e Biosistemi, 00136 Rome, Italy; sofia.giorgetti@unipv.it (S.G.); v.bellotti@ucl.ac.uk (V.B.); 3Department of Molecular Medicine, Institute of Biochemistry, University of Pavia, 27100 Pavia, Italy; chiara.mimmi@unipv.it (M.C.M.); palma.mangione@unipv.it (P.P.M.); 4Wolfson Drug Discovery Unit, Centre for Amyloidosis and Acute Phase Proteins, Division of Medicine, University College London, London NW3 2PF, UK; g.verona@ucl.ac.uk (G.V.); graham.taylor@ucl.ac.uk (G.W.T.); 5Scientific Direction, Fondazione IRCCS Policlinico San Matteo, 27100 Pavia, Italy

**Keywords:** TTR, amyloidosis, mechano-enzymatic mechanism, microcalcification, calcium dysregulation

## Abstract

The plasma protein transthyretin (TTR), a transporter for thyroid hormones and retinol in plasma and cerebrospinal fluid, is responsible for the second most common type of systemic (ATTR) amyloidosis either in its wild type form or as a result of destabilizing genetic mutations that increase its aggregation propensity. The association between free calcium ions (Ca^2+^) and TTR is still debated, although recent work seems to suggest that calcium induces structural destabilization of TTR and promotes its aggregation at non-physiological low pH in vitro. We apply high-resolution NMR spectroscopy to investigate calcium binding to TTR showing the formation of labile interactions, which leave the native structure of TTR substantially unaltered. The effect of calcium binding on TTR-enhanced aggregation is also assessed at physiological pH through the mechano-enzymatic mechanism. Our results indicate that, even if the binding is weak, about 7% of TTR is likely to be Ca^2+^-bound in vivo and therefore more aggregation prone as we have shown that this interaction is able to increase the protein susceptibility to the proteolytic cleavage that leads to aggregation at physiological pH. These events, even if involving a minority of circulating TTR, may be relevant for ATTR, a pathology that takes several decades to develop.

## 1. Introduction

Transthyretin (TTR) is a protein present in plasma and cerebrospinal fluid (CSF), which acts as a transporter of the thyroid hormone thyroxine (T4). It is also involved in the transport of retinol through the interaction with retinol binding protein (RBP). TTR has been recognized as the precursor of systemic amyloidosis both in its wild type form, responsible for wild type TTR amyloidosis (ATTRwt), which affects 10–25% of people over 80 years of age [1], and in its more than 100 variants causing hereditary TTR amyloidosis (ATTRv). A possible protective role of TTR in Alzheimer’s disease has also been postulated [2]. In addition, TTR can bind a range of different ligands and can therefore act as a ligand reservoir in the same way as albumin. To date, a broader functional role for TTR cannot be excluded. The mechanism underlying the conversion of soluble globular TTR into insoluble amyloid fibrils has undergone extensive investigation starting from the seminal observation that the full-length protein can form aggregates in acidic conditions [3]. Aggregates formed under these conditions comprise full-length TTR only, although analysis of the biochemical composition of ex vivo TTR amyloid fibrils has highlighted, in most cases, the presence of both full-length and fragmented TTR in the deposits [4].

The investigation of the role of proteolysis in TTR amyloidogenesis has more recently led to the identification of a mechano-enzymatic mechanism, characterized by the synergic actions of proteolysis and shear stress, through which genuine amyloid fibrils, comprising both full-length and fragmented TTR, can be generated in a more physiological environment [5,6,7]. Both these in vitro methods have indicated that tetramer disassembly is a crucial step in TTR amyloidogenesis. Tetramer dissociation can be favored by specific destabilizing mutations but what drives or favors the tetramer disassembly, in particular for wild type TTR, is still not completely clear. It has recently been suggested that calcium, whose homeostasis is often dysregulated in the aged population, might play a role in the destabilization of the structure of TTR [8]. In the body, nearly 99% of total calcium is immobilized in the bones as calcium hydroxyapatite and only about 1% is available in the extracellular fluid where its concentration varies between 2.1 and 2.6 mM [9]. Approximately 40% of extracellular calcium is bound to proteins, predominantly to human serum albumin (HSA), 10% is complexed with bicarbonate, citrate, phosphate and lactate anions [10] and that remaining is unbound (~1.3–1.5 mM). It has been known for over 60 years that the TTR fraction of human serum binds calcium with a similar affinity to albumin [11]. The apparent binding constant for HSA–calcium is about 0.6 mM [12] with four similar low-affinity binding sites. Since the initial observations in 1959, little work has been carried out on TTR–calcium binding. Recently, the X-ray structures of wild type TTR in the presence of 200 mM CaCl_2_ (pdb: 4mrb [5] and 4n85 [13]) revealed the presence of one or two Ca^2+^ ions bound/dimer in the crystals. On the contrary, a number of different analytical techniques failed to reveal any specific interaction between calcium and TTR in solution [8,13]. The work carried out by Wieczorek and collaborators [8] using a variety of analytical techniques suggests that calcium in the millimolar range destabilizes TTR structure by altering its conformational flexibility towards non-native conformers. Whether calcium actually binds to TTR in solution still remains unclear.

Here, we have applied high-resolution NMR spectroscopy to investigate calcium binding to TTR and we have assessed its effect on TTR aggregation at physiological pH in the presence of both proteolysis and mechanical forces.

## 2. Materials and Methods

### 2.1. Protein Expression and Purification

Transformed BL21 star (DE3) cells (Thermo Fisher Scientific, Waltham, MA, USA), containing the peTM11 plasmid encoding hexahistidine-tagged wild type TTR, were plated onto Luria broth (LB)–agar media containing 30 µg/mL kanamycin for overnight incubation at 37 °C. A single colony was isolated and cultured overnight at 37 °C in 5 mL LB medium containing 30 μg/mL kanamycin under shaking conditions (LB/kan). This preparation was inoculated into 1 L LB/kan for an initial growth at 37 °C. When the culture reached OD_600_ = 0.5, the temperature was reduced to 30 °C. Protein expression was induced at OD_600_ = 0.6 by adding IPTG (1 mM final concentration) for overnight incubation. The cells were harvested by centrifugation at 3500× *g*, suspended in lysis buffer containing 20 mM Tris-HCl pH 8, 250 mM NaCl, 3 mM imidazole and finally sonicated at 4 °C. The supernatant was clarified following 30 min centrifugation at 18,000× *g* and loaded onto a HisTrap FF crude nickel affinity chromatography column (GE Healthcare, Chicago, IL, USA) equilibrated in lysis buffer. After extensive washing with 20 mM Tris-HCl, 10 mM imidazole, containing stepwise increasing concentrations of NaCl (250 mM, 500 mM and 1 M, respectively), the protein was eluted with 20 mM Tris-HCl, 250 mM NaCl, 250 mM imidazole, pH 8.0. His-tagged TEV protease (Sigma-Aldrich, St. Louis, MO, USA) was added at 1% *w*/*w* during dialysis to selectively cleave the hexaHistidine-tag, which was then removed by affinity chromatography, together with the enzyme. Fractions containing TTR were pooled and subjected to size exclusion chromatography using a Superdex 75 Hi Load 26/60 column (GE Healthcare, Chicago, IL, USA) equilibrated and eluted with 25 mM Tris-HCl, 100 mM NaCl, pH 8.0. Fractions containing TTR were dialyzed against water at 4 °C for at least 3 days and then lyophilized. Purity and molecular weight were determined by SDS-PAGE analysis and mass spectrometry, respectively. For the expression of recombinant [75% ^2^H,^13^C,^15^N] wild type TTR, the initial overnight culture was adapted to grow in a deuterated background by stepwise addition of Ross medium prepared in ^2^H_2_O supplemented with ^15^N ammonium sulfate and with ^13^C glucose for 7 steps and further growth steps were carried out in the same medium.

### 2.2. NMR Spectroscopy

NMR spectra were obtained at 700 MHz with a Bruker AVANCE NEO (Bruker, Billerica, MA, USA) on 80–125 μM U-[^2^H,^13^C,^15^N] recombinant TTR samples in 10 mM HEPES buffer at pH 6.5 or 7.4 with and without 154 mM NaCl, in a mixture of H_2_O/D_2_O 95/5 *v*/*v*. A reference 2D [^15^N, ^1^H] TROSY spectrum [14] for apo-TTR was acquired at 25 °C, together with additional spectra at 28, 31, 34, 37 °C, respectively. The spectra at various temperatures were necessary to obtain the assignment at 25 °C from the already known assignment at 37 °C. The 2D [^15^N, ^1^H] TROSY spectra were acquired upon addition of small aliquots of stock CaCl_2_ solutions 0.025–2 M with a maximum volume variation of 1.5%.

The 3D HNCA spectra were acquired at 298 K to check the assignment at the final CaCl_2_ concentration.

The chemical shift deviation of individual amide pairs, Δδ, was defined as [15]:$$\Delta \delta =\sqrt{{\left(\Delta \delta_{HN}\right)}^{2}+{\left(\frac{\Delta \delta_{N}}{6.5}\right)}^{2}}.$$

Spectra were processed with Topspin 4.0.9 (Bruker Biospin, Billerica, MA, USA) and analyzed in NMRFAM-SPARKY [16]. The non-linear regression to obtain the dissociation constants was performed with Mathematica 11.

### 2.3. Electrostatic Calculation

The electrostatic profile of TTR was determined starting from the X-ray structure refined at 1.7 Å resolution (pdb: 1tta [17]) for the core region 10–124, that also contains the N-ter G1-K9 and C-ter P125-E127 regions modeled by simulated annealing. The presence of the two charged terminals of K9, K126 and E127, is essential for a correct evaluation of the overall electrostatic potential. APBS software [18] was used to calculate the electrostatic potential after the necessary addition of hydrogens with VMD tools [19] and after the correct protonation state of histidine residues at neutral pH. We found that H31, H56 and H88 are fully protonated and H90 is protonated in the ε position. The electrostatic potential was visualized on the protein surface with Pymol.

### 2.4. Molecular Dynamics (MD) Simulations

We used as the starting configuration the X-ray structure taken from the RCSB Protein Data Bank (pdb) [20] with id 5cn3, solved at 1.30 Å [21]. The first 9 N-terminal residues and the last 2 C-terminal residues are not present in the structure and we did not model them. The integrity of the crystal structure was checked with Molprobity online software [22] and no missing backbone or side chain atoms were found. With the pdb_delhetatm tool [23], we removed crystallization water molecules and with the pdb2pqr tool [24] we determined the protonation state of histidine residues at pH 7.4. These protonation states were assigned to the structure with VMD [19].

Nanoscale Molecular Dynamics (NAMD) [25] software was used to perform the simulations employing Chemistry at Harvard Macromolecular Mechanics (CHARMM) all atoms force field [26]. We first minimized the prepared structure in vacuum for 500 steps. Then, we placed the minimized structure in the center of a cubic box of 10 × 10 × 10 Å^3^ and the system was solvated with a 3-site rigid water model (TIP3P) [27] and neutralized with 0.1 M NaCl. The resulting protein concentration was approximately 3 mM. CaCl_2_ was added in order to obtain the same protein/Ca^2+^ ratio reached at the end of the NMR titration carried out at pH 7.4 in the presence of NaCl. We minimized and equilibrated water to the target temperature of 298 K, keeping the protein and the ions fixed. The minimization was performed for 500 steps and the equilibration run for 20 ps. Then, the same two steps were carried out for the whole system without constraints with an equilibration step of 120 ps. The final production run was performed under NPT conditions for 15 ns, saving coordinates after every ps for analysis. Calcium atoms within 3.5 Å of the protein were identified as interacting ions [28]. The distances between the protein residues and Ca^2+^ were measured over all the trajectory frames to infer the interaction persistency during simulation time.

### 2.5. Aggregation

Proteolysis-mediated fibrillogenesis of wild type TTR was carried out in glass vials (air/water interface of 1.5 cm^2^) stirred at 1500 rpm (IKA magnetic stirrer) and at 37 °C for 96 h using 1 mg/mL TTR (corresponding to 18 µM tetrameric TTR) in 10 mM HEPES, 154 mM NaCl, pH 7.4 in the presence of 0 mM, 1 mM, 10 mM, 40 mM and 60 mM CaCl_2_, respectively. Proteolysis was initiated by addition of trypsin (5 ng/µL) to yield a final protease to TTR ratio of 1:200 *w*/*w*. Alteration of trypsin activity due to high CaCl_2_ concentration was excluded by testing D-VLK 4-nitroanilide HCl peptide hydrolysis at 310 K using a spectrophotometric assay at 405 nm with and without 60 mM CaCl_2_. A control sample subjected to agitation only without enzyme was also included. At the end of the incubation period, aggregation was monitored by light scattering at 400 nm using a Jasco V650 spectrophotometer. Each sample was then centrifuged for 20 min at 10,300× *g*, supernatant was removed and the pellet was washed twice in PBS before being suspended in 100 µL of 10 µM thioflavin T (ThT) in PBS, pH 7.4 for further investigation by ThT emission fluorescence at 480 nm, following excitation at 445 nm, using a FLUOstar Omega plate reader (BMG Labtech, Ortenberg, Germany). A 10 µL aliquot of pellet was stained with alcoholic alkaline Congo red (CR) solution and the pathognomonic amyloid birefringence was observed with polarized light microscopy. Samples were also analyzed by SDS–gradient 4–15% PAGE (BioRad, Hercules, CA, USA) under reducing conditions in order to collect information on protein consumption and formation of amyloidogenic fragments. Statistical analysis on both turbidity and ThT aggregation data was performed using the Friedman test (non-parametric analogue to one-way ANOVA for repeated measures).

## 3. Results

### 3.1. Electrostatics of TTR

Since our study concerns the interaction between two charged species, i.e., Ca^2+^ and TTR, we preliminarily assessed the electrostatic properties of TTR. Based on electrostatic calculations, TTR is negatively charged at neutral pH with a pI = 5.35 and a net charge of −8.27 (Appendix A). The electrostatic potential represented on the Connolly surface evidences a moderately positive surface in the frontal view parallel to the channel connecting the thyroxine, T4, binding pockets (Figure 1A), a more positive patch at the entrance of the binding site where the negatively charged T4 enters (Figure 1B) and a large negative region at the apical ends (Figure 1C). Furthermore, a closer look reveals a strongly negatively charged cavity formed by E61, E62, E63, E66 and D99 (Figure 1D) characterized by a prominent negative isopotential surface at −1.0 kT/e with a corresponding positive isopotential surface at 1.0 kT/e much closer to the protein surface (Figure 1E).

### 3.2. Does TTR Bind Ca^2+^?

The 2D [^15^N, ^1^H] NMR correlation maps provide a very informative fingerprint of a protein, including atomic-level information about the NHs of all residues of a polypeptide chain, except prolines. The chemical shift of each cross-peak is extremely sensitive to the chemical environment and to residue conformation, making the chemical shift perturbation (CSP) analysis a very powerful tool for studying protein–ligand interactions. The changes in peak chemical shift (Δ*δ*) are also diagnostic of the rate of the exchange processes, which are classified in NMR based on the deviation of a peak from its reference position. Interacting proteins and ligands are in fast exchange when k_ex_ = k_on_ + k_off_ >> Δ*δ* and the shift of an exchanging peak is the population-weighted average between the bound and free forms. When Δ*δ* >> k_ex_, slow exchange occurs and the intensity of the free peak gradually decreases, while the intensity of the bound peak correspondingly increases, linearly with the concentration of the free and bound proteins. In our study, we use the unique capabilities of NMR to investigate, at the residue level, the interaction of TTR with CaCl_2_ in solution.

TTR is a 55 kDa protein characterized by a correlation time of about 20 ns at 25 °C, estimated from its molecular weight [29]. Such a relatively slow global rotational motion requires the use of specific strategies to overcome the fast signal decay: (1) perdeuteration [30] and (2) use of TROSY sequences [14]. The 2D [^15^N, ^1^H] TROSY spectrum of triple labeled [75% ^2^H, ^13^C, ^15^N] apo-TTR acquired at 16.44 T was previously assigned at 37 °C [31] and the assignment was transferred to 25 °C by tracking the chemical shift deviation of spectra acquired every 3 °C.

Small volumes of stock CaCl_2_ solutions were added to a TTR solution in HEPES buffer and in different salt and pH conditions, to a final CaCl_2_ concentration of 60–100 mM. A series of superimposed 2D TROSY spectra, in Figure 2 and Appendix A, acquired at 25 °C with different Ca^2+^ concentrations, show that TTR and Ca^2+^ are in a fast exchange regime as the peaks progressively shift their positions during titration.

Δ*δ_obs_* values calculated relative to apo-TTR are reported in the bar plots of Figure 3A,C,E which illustrate a series of peaks that significantly change their chemical shift upon Ca^2+^ addition. The Δ*δ*s of TTR NHs in different experimental conditions at a comparable Ca^2+^/TTR ratio are reported in the X-ray structure (pdb: 5cn3) as a color gradient from yellow to green in Figure 3B,D,F.

The interpretation of our CSP data, suggesting the involvement of different TTR regions in the Ca^2+^/protein interaction, can be supported by the analysis of literature X-ray data. The crystal structure (pdb: 4mrb) of TTR crystallized in a buffer containing 200 mM CaCl_2_ shows a Ca^2+^ binding site (site-1, Figure 4A) formed by the backbone carbonyl and the carboxylic group of D99 and by the carboxylic group of E66 along with a water molecule (not shown). Another X-ray structure of TTR with Ca^2+^ (pdb: 4n85) shows, in addition to site-1, for both chains, a Ca^2+^ ion bound to C10 sulfur and N*δ* of the H56 imidazole ring (site-2, Figure 4C). Based on these observations, we analyzed the CSP data at pH 7.4 (without NaCl) and searched for NMR data compatible with site-1 and -2. Indeed, D99 is the most shifting peak and E66 shifts significantly indicating that Ca^2+^ also binds to site-1 in solution.

Site-2, on the other hand, is not clearly detected as neither C10 nor H56 were observed in 2D TROSY at neutral pH, but a clue comes from R104 and K126, which have a Δ*δ* ≳ average +2σ and are spatially close to site-2. The observation of C10 amide at neutral pH is prevented by the exchange with water, but the peak becomes visible when the pH is lowered. We therefore repeated the titration with CaCl_2_ at pH 6.5, and indeed C10 is observed and shifts significantly with the addition of Ca^2+^ (Figure 3A). As before, we could not detect H56 NH under our experimental conditions. The CSP of R104 can be explained by the formation of a hydrogen bond (Hb) that occurs between C10-O and R104-Nε upon Ca^2+^ binding (Figure 4C) and is absent without Ca^2+^ (Appendix A). In addition, the R104 side chain adopts different orientations in the presence and in the absence of Ca^2+^, which affect the opposite amide nitrogen of K126 that actually shifts during titration (Appendix A).

To determine the dissociation constants of the binding of Ca^2+^ to site-1 and -2, CSP data were analyzed using the $TTR+Ca^{2+}⇌TTR-Ca$ model. The observed chemical shift, *δ_obs_*, can be expressed as $\delta_{obs}=\delta_{free}\chi_{free}+\delta_{bound}\chi_{bound}$ [32], where *δ* and *χ* are the chemical shift and molar fraction of the free and bound forms, as indicated by the subscript. The dissociation constant, *K_d_*, can thus be obtained by a non-linear interpolation of Equation (1):(1)$$\Delta \delta_{obs}=\Delta \delta_{max}\left\{\frac{{\left[TTR\right]}_{0}+{\left[Ca^{2+}\right]}_{0}+K_{d}-\sqrt{{\left({\left[TTR\right]}_{0}+{\left[Ca^{2+}\;\right]}_{0}+K_{d}\right)}^{2}-4{\left[TTR\right]}_{0}{\left[Ca^{2+}\right]}_{0}}}{2{\left[TTR\right]}_{0}}\right\}$$
where [*TTR*]_0_ and [*Ca*^2+^]_0_ are the total concentrations of monomeric TTR and Ca^2+^, respectively, and Δ*δ_max_* is the maximum chemical shift at saturation. The Δ*δ_obs_* of NHs belonging to the same binding site were fitted as a system of equations. At pH 6.5, the fitting of CSP data gives for binding site-1 *K_d_* = (8.7 ± 2.1) mM and for binding site-2 *K_d_* = (13.5 ± 1.3) mM (Figure 4B,C).

To verify the binding of Ca^2+^ to TTR at physiological pH and salt concentration, 2D TROSY at different Ca^2+^/TTR ratios were also acquired at pH 7.4 and in the presence of 154 mM NaCl. The results show that binding still occurs but the CSP is less pronounced than the one measured without NaCl (Figure 3A,C,E). At high salt concentration, the binding is affected by electrostatic shielding and under these conditions we obtained *K_d_* = (32.7 ± 4.3) mM for site-1 (Figure 4C). For site-2, we could not observe any peak corresponding to directly bound residues (H56 and C10), but we estimated the *K_d_* by fitting the chemical shift variation of R104 that is indirectly involved in the interaction with the cation. By fitting R104 CSP (Figure 4F), we obtained *K_d_* = (40.8 ± 6.9) mM.

From CSP data, other residues that shift by more than the average + 2σ are the negatively charged E51 in loop CD (site-3) and E62 in the loop DE (site-4), which are exposed to the solvent and do not participate in any H bonding. Moreover, E62 is also part of the negatively charged cavity described in Section 3.1. According to CSP analysis, two additional putative binding sites comprise K76 (helix H1) and E89 (loop H1-F) (site-5 in Appendix A), which are connected by two Hbs, and V14-K15-A25-G53-E54-L55-A109, which are involved in a complex Hb network (site-6 in Appendix A). The *K_d_* values (Table 1) for these putative binding sites, which were not observed in the X-ray data, were also determined (Appendix A) and are in a similar range as the one measured for site-2.

### 3.3. Binding Specificity

Chemical shift differences in different solvents are usually described by four terms: bulk susceptibility of the solvent, electrostatic screening, van der Waals forces between the protein and the solvent and, finally, hydrogen bond formation [33]. In the case of NaCl, the effect has been mainly described as a consequence of a long-range bulk susceptibility where polarized water produces a small magnetic field with a screening effect on all nuclei with an average chemical shift variation on H^N^ (Δ*δ*^H^) of −0.01 ± 0.01 ppm per 100 mM NaCl [33] as probed in hen egg white lysozyme (HEWL). To rule out the possibility that the effect we observed with CaCl_2_ was due to a generic salt effect, we acquired TTR spectra at different NaCl concentrations. Our results report an average Δ*δ*^H^ = −0.006 ± 0.025 ppm at 100 mM NaCl (Appendix A) with a clear indication of a less uniform variation in chemical shift than that reported in the literature [33] (for comparison, at 100 mM CaCl_2_ the average Δ*δ*^H^ = −0.022 ± 0.036 ppm). It is worth noting that at physiological pH, TTR is a negatively charged protein while HEWL is positively charged (pI = 9.39), and that the leading effect of Cl^−^, according to the Hofmeister series, may therefore be more pronounced for positively charged proteins. Furthermore, NaCl maps the electrostatic negative regions of TTR similarly to CaCl_2_ indicating that with both salts a solvent bulk effect can be ruled out. The comparison between CaCl_2_ and NaCl data highlights that among the six putative binding sites identified, only binding site-1 is specific for Ca^2+^ while the others may indicate sites of localized persistent electrostatic interactions.

### 3.4. Molecular Dynamics Simulations

MD simulations run for 15 ns in the presence of CaCl_2_ evidenced that Ca^2+^ clusters in specific regions. To infer the persistency of the interaction, the distances between interacting residues and calcium ions were measured and the percentage of configurations showing distances ≤3.5 Å over all the trajectory is plotted in Appendix A. We thus identified that D38, D39, E42, E51, E54, E61, E62, E63, E66, E72, E89, E92 and D99 interact with Ca^2+^ for more than 50% of the simulation time. These residues can be grouped in three negative patches: the apical region including D38, D39, E42, E72, and E92; the cavity made by E61, E62, E63, E66 and D99; and the protruding region that comprises E51 and E54 (Appendix A). It is worth noting that D99 is the only residue that during the simulation interacts with calcium not only with its side chain carboxyl, but also with its backbone carbonyl.

### 3.5. Ca^2+^ Favors TTR Aggregation under Physiological pH

TTR was dissolved in HEPES, pH 7.4 and incubated for 96 h in the presence of different concentrations of CaCl_2_ under mechano-enzymatic conditions of fibrillogenesis [5,6]. Aggregation was monitored by both generic light scattering at 400 nm and amyloid-specific ThT emission fluorescence at 480 nm (Figure 5A,B). Good agreement was observed between the two data sets with calcium binding favoring TTR fibrillogenesis at higher concentrations (40 mM and 60 mM), while no significant effect was observed at lower Ca^2+^ concentrations (1 mM and 10 mM) with respect to the Ca^2+^-free control sample. The median equality of absorbance at 400 nm/ThT emission fluorescence at different levels of CaCl_2_ concentration was verified using the Friedman test. The two parameters were measured in the same samples before and after subsequent addition of CaCl_2_ to obtain [Ca^2+^] = 0, 1, 10, 40, 60 mM. The *p*-value associated with turbidity at 400 nm (0.03875) was smaller than the fixed threshold, a = 0.05, while the *p*-value associated with ThT was not (0.4337). In spite of the intrinsic high data dispersion and the small sample size, these data indicate that fibrillogenesis is favored by high Ca^2+^ concentration. Previous work by our group [6] has shown that in vitro wild type TTR can generate genuine amyloid fibrils, comprising a mixture of both full-length TTR and the amyloidogenic 49–127 C-terminal fragment. The investigation of the aggregated material by SDS-PAGE highlighted an increased formation of aggregation-prone 49–127 TTR fragments in the presence of higher Ca^2+^ concentrations (Figure 5C), which explains the increased rate of aggregation observed. The pathognomonic birefringence under cross-polarized light following Congo red staining was used to confirm the presence of amyloid fibrils in all samples subjected to trypsin digestion.

## 4. Discussion

A number of metal cations, such as Zn^2+^, Mn^2+^, Cu^1+,2+^ and Fe^2+,3+^, show the formation of a complex with TTR when soaked in crystallization media with high metal concentrations. Zn^2+^ and Cu^2+^ also promote amyloid formation, especially in the L55P variant TTR in the presence of Zn^2+^ [34]. In addition, the protective role of TTR in Aβ fibril formation and the presence of metal ions in Aβ amyloid deposits have contributed to increasing the interest in studying the interaction between TTR and metal ions [35,36,37,38]. Crystallographic structures show that Ca^2+^ is located at two binding sites, the first formed by E66 and D99 and the second by H56 and C10, located respectively in loop DE, loop FG, loop DE and N-terminus of strand A, without involving the T4 binding pockets. Recently, Wieczorek and colleagues published a very detailed analysis of TTR structural stability in the presence of Ca^2+^ without providing evidence of binding in solution, but showing several effects that can be associated with changes in the hydration shell that could indirectly affect TTR dynamics. An increase in aggregation propensity under non-physiological acidic conditions was also reported [8]. In this study, we show that Ca^2+^ binds with low affinity to TTR in solution without affecting its native structure and that the binding is ionic strength dependent. Indeed, the chemical shift variations of the amides in the 2D TROSY spectra indicate that the TTR structure remains essentially unchanged upon binding. Ca^2+^ also binds to site-1, comprising D99 and E66, in solution with an affinity in the mM range while the other sites may represent local electrostatic interactions as high NaCl concentrations also induce similar spectra variations. The localized effect of NaCl and CaCl_2_ mainly in negatively charged TTR regions rules out a generic effect of salts on water in line with other authors that propose a direct interaction between ions and macromolecules in spite of the creation/breaking of bulk water structures [39]. More interestingly, the experiments with NaCl point out that only site-1 is specific for Ca^2+^ while the other sites likely report only of electrostatic interactions without a specific binding site. NMR and electrostatic data together with MD simulations suggest that Ca^2+^ clusters in close proximity to negatively charged patches and only after exploring the protein surface does the cation bind to site-1 with a mM affinity. These data agree with previous MD simulation studies reporting that calcium, before reaching its binding site on calbindin, is detained within 3.5 Å of the protein surface for a long time (approximately 650 ns) during which the cation scans the surface searching for higher affinity binding sites. This specific mechanism appears to be particularly important at low Ca^2+^ concentrations [28]. If electrostatics play a crucial role in Ca^2+^ binding, we should keep in mind that the screening effect of NaCl, which causes the 4-fold increase in *K_d_* measured in vitro with 154 mM NaCl, may be reduced in vivo. In the extracellular matrix, where aggregation takes place, the presence of glycosaminoglycans (GAGs) buffers Na^+^, which in the interstitial fluid bridges GAGs and modulates their polymerization grade, thus reducing free Na^+^ concentration [40]. Ca^2+^ affinity close to the protein surface can be modulated in vivo by different players, as aforementioned, but the investigation of these details is outside the aim of our study.

Despite small structural variations, TTR aggregation propensity at physiological pH is favored by Ca^2+^ addition, similarly to the previously reported observations obtained under acidic conditions [8]. Moreover, high Ca^2+^ concentrations favor the proteolytic cleavage with formation of the aggregation-prone 49–127 TTR fragment. Normal extracellular calcium concentration averages at 2.4 mM [9] and can be further locally modulated in terms of distances from interfaces [41]. Considering an extracellular concentration of calcium around 2.4 mM and the *K_d_* values obtained by NMR titration experiments, we can estimate that the percentage of Ca^2+^-bound TTR in vivo is about 7%. Albeit modest, these values indicate that a non-negligible percentage of TTR is bound to Ca^2+^ in vivo, thus increasing the susceptibility of a fraction of circulating TTR to serine protease cleavage. In addition, there is recent evidence that microcalcification is a common feature in cardiac tissue of patients affected by cardiac TTR amyloidosis [42], thus suggesting that local calcium concentration in the extracellular space of the heart, where amyloid deposition takes place, may indeed vary. The microcalcifications have a calcium content per gram of tissue about 40,000 times higher than in microcalcification-free areas [43], making the hypothesis of local high calcium content reasonable. Amyloid fibrils containing a mixture of both full-length and fragmented TTR have also been identified in other tissues, such as the vitreous body [4], although whether calcium might be involved in promoting aggregation is difficult to postulate as not enough information about its local concentration is available for these patients. Our findings still leave important questions with clinical relevance open: in addition to a pre-formed TTR–Ca^2+^ complex more prone to fibril formation, is Ca^2+^ able to stabilize aggregation prone partially folded species and/or does an already formed fibril tend to bind calcium and, by doing so, increase the propensity for fibril elongation? An answer to these questions could also be relevant for scintigraphy that is used as a crucial diagnostic tool in the clinic, in which the association of 99mTechnetium-3,3-diphosphono-1,2-propanodicarboxylic acid (99mTc-DPD) with amyloid plaques is described as calcium dependent, and in which the presence of microcalcifications in cardiac ATTR appears to be related to DPD specificity for this type of amyloidosis [44].

## Figures and Tables

**Figure 1 biomolecules-12-01066-f001:**
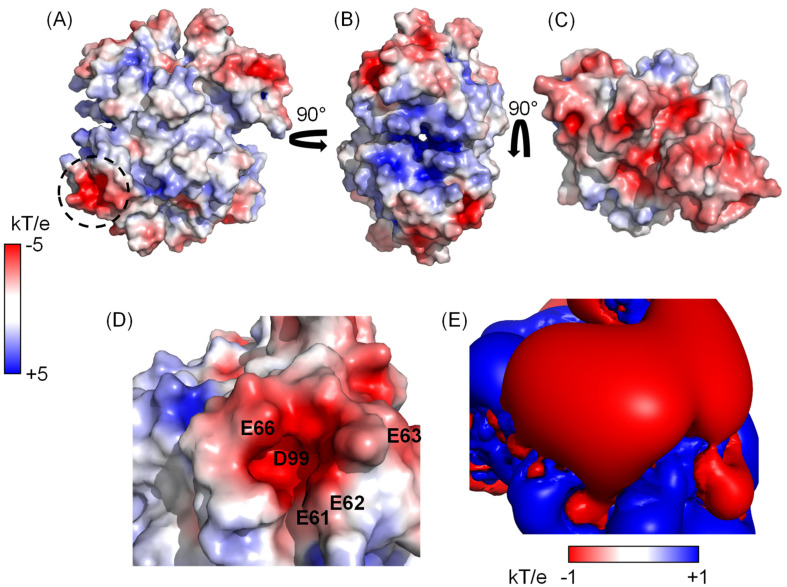
Electrostatic potential mapped on the surface of TTR (pdb: 1tta). In (**A**–**C**), the whole tetrameric structure with different orientations is depicted, while in (**D**) the highly negatively charged cavity formed by E61, E62, E63, E66 and D99 is zoomed into and the corresponding ±1 kT/e isopotential electrostatic field is represented in (**E**). The dashed circle in panel (**A**) corresponds to the zoomed in region of panels (**D**,**E**). All the calculations were performed using APBS [18] in Pymol.

**Figure 2 biomolecules-12-01066-f002:**
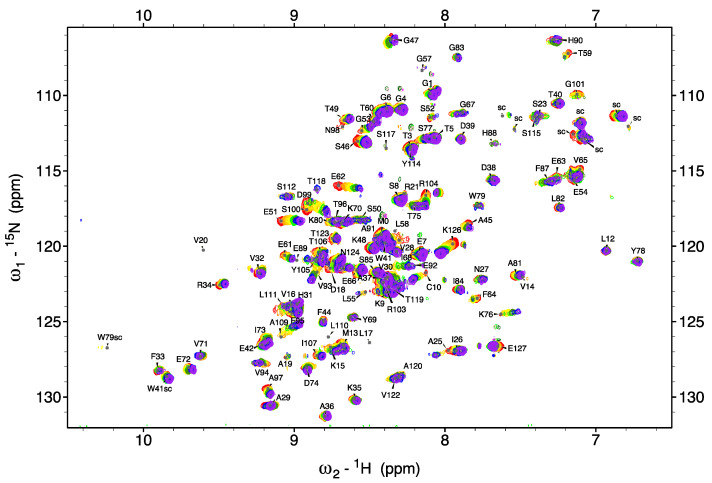
Overlay of 2D [^1^H, ^15^N] TROSY spectra of TTR acquired at 700 MHz recorded at increasing calcium concentration at pH 6.5. The color code is: 0 mM red, 5 mM orange, 10 mM gold, 20 mM yellow, 40 mM green, 60 mM chartreuse green, 80 mM blue and 100 mM purple.

**Figure 3 biomolecules-12-01066-f003:**
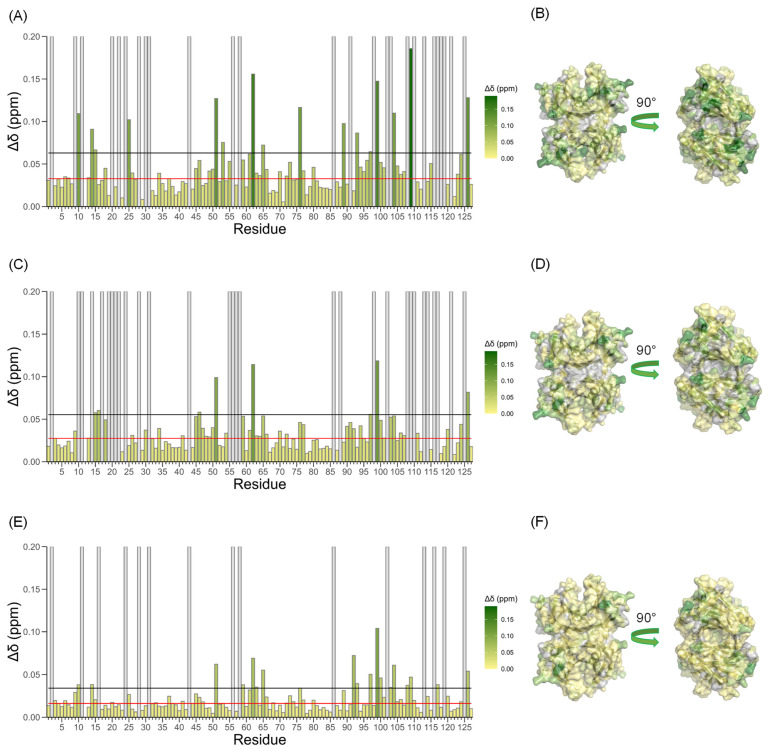
Bar plot of combined NH chemical shift variation recorded at TTR/Ca^2+^ = 0.2 at pH 6.5 (**A**), at pH 7.4 (**C**) and at pH 7.4 in the presence of 154 mM NaCl (**E**). Gray bars indicate prolines and residues that could not be followed during titration. The red lines correspond to the average value and the black ones to the average +2σ. Chemical shift perturbation (Δ*δ*) observed at pH 6.5 (**B**), at pH 7.4 (**D**) and at pH 7.4 with 154 mM NaCl (**F**) in the presence of CaCl_2_ shown in the structure of TTR (pdb: 5cn3) with the same color gradient as in the bar plots.

**Figure 4 biomolecules-12-01066-f004:**
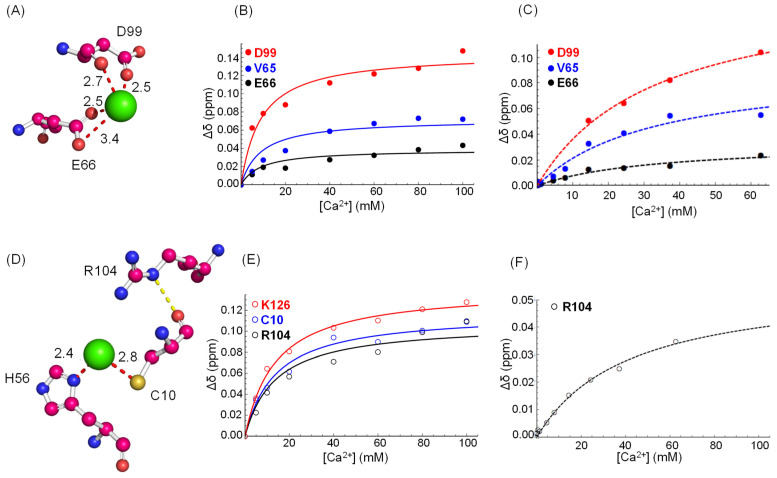
Ca^2+^ binding sites shown by 4mrb (**A**) and 4n85 (**D**) X-ray structures. Chemical shift variation (Δ*δ*) as a function of Ca^2+^ concentration for the peaks belonging to binding site-1 at pH 6.5 (**B**) and at pH 7.4 with 154 mM NaCl (**C**) and for binding site-2 at pH 6.5 (**E**) and at pH 7.4 with 154 mM NaCl (**F**). The lines correspond to the data fitting by Equation (1).

**Figure 5 biomolecules-12-01066-f005:**
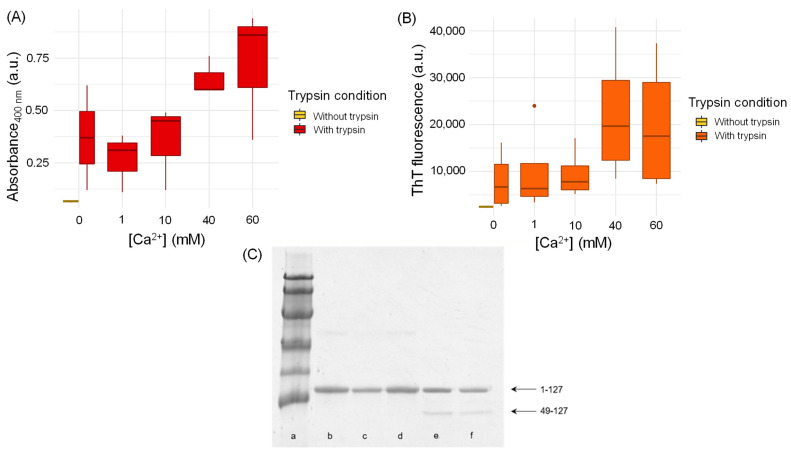
Effect of calcium on mechano-enzymatic wild type TTR fibrillogenesis. (**A**) Light scattering signal at 400 nm monitored in the whole sample after 96 h of aggregation. (**B**) ThT emission fluorescence at 480 nm following excitation at 445 nm measured in pellets after 96 h of aggregation. Statistical analysis on both turbidity and ThT aggregation data was performed using the Friedman test. (**C**) SDS 15% PAGE under reducing conditions. Lane a: marker proteins (14.4, 20.1, 30.0, 45.0, 66.0 and 97.0 kDa); lane b: WT TTR; lane c: WT TTR 1 mM Ca^2+^; lane d: WT TTR 10 mM Ca^2+^; lane e: WT TTR 40 mM Ca^2+^; lane f: WT TTR 60 mM Ca^2+^. 1–127 and 49–127 indicate full-length TTR and the amyloidogenic C-terminal fragment, respectively.

**Table 1 biomolecules-12-01066-t001:** CSP fitting parameters obtained using Equation (1) for the binding sites inferred from NMR data.

Site	*K_d_* (mM)	R^2^; *p*-Value
Site-1	8.7 ± 2.1	0.993; 6.3 × 10^−3^
Site-2	13.5 ± 1.3	0.999; 5.0 × 10^−5^
Site-3	16.1 ± 2.2	0.998; 3.3 × 10^−4^
Site-4	16.2 ± 3.8	0.993; 5.4 × 10^−3^
Site-5	19.5 ± 6.2	0.973; 6.9 × 10^−3^
Site-6	15.4 ± 2.1	0.979; 2.4 × 10^−9^

## Data Availability

Not applicable.

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
