# Peer review of "Calcium Binds to Transthyretin with Low Affinity"

_biomolecules, 2022, doi:10.3390/biom12081066_

Round 1
Reviewer 1 Report
The manuscript “Calcium binds to transthyretin with low affinity” (manuscript 1846284) by Cantarutti, et al. examines Ca2+ binding to the transthyretin (TTR) protein. Nuclear Magnetic Resonance spectroscopy (NMR) measurements on recombinant TTR protein are made under multiple temperatures and with various Ca2+ concentrations, some in the presence of Na+ ions. This data was supplemented with calculations of electrostatic properties of the protein, molecular dynamics simulations and measurements of TTR aggregation by light scattering and gel electrophoresis. In general, this manuscript is well written and provides sufficient detail on the experimental methods used. The experiments are generally well designed and most of the overall conclusions are supported by the results presented. The data presented is internally consistent and appropriate controls are provided for most experiments. There are some concerns about the manuscript in its current form:
1) The vast majority of these experiments are conducted at super-physiological levels of Ca2+ (5-100mM) and the changes observed in the protein are not revealed until well into that range. There is little discussion about the implications of that for the interpretation of this work, especially from the perspective of the role that Ca2+ binding at these concentration might have in vivo. While there is some of this in the Discussion (line 403) this could require a more critical eye in the Discussion and some mention in the Abstract.
2) It would be helpful to supplement the data in Figure 4C and 4D (at pH 7.4) with additional panels for the results at the same residues under the pH 6.5 conditions should be presented in the main figures.
3) Line 16: It should be made clear in the abstract how TTR is responsible for the second most common form of systemic (ATTR) amyloidosis. This could be done by mentioning that changes in expression level or genetic mutation of TTR lead to ATTR.
4) Line 42, 369: There should be a paragraph break here.
5) Line 57: The sentence “Calcium homeostasis involves…” is unnecessary information and could be removed from the introduction.
Author Response
"Please see the attachment."

Reviewer 2 Report
Cantarutti et al. studied calcium binding to transerythrin. Technically, the study is performed carefully. Using and interesting combination of approaches and methods, the authors were able to overcome methodological challenges (size of the protein investigated by NMR) obtain clear results.
As the authors admit, biological relevance of the findings is less clear. Therefore, it is difficult to judge significance of the results and interest to the readers. There is also a certain inconsistency in presenting the results. When studying TTR aggregation, the authors use pH 7.4 and 154mM NaCl as relevant conditions. However, when presenting determination of dissociation constants, results obtained at pH 6.5 and no additional NaCl are presented in the main text and Fig. 4, arguing in the discussion that the relevant sodium concentration is reduced by the presence of glycosaminoglycans. The authors should present all data obtained at the same conditions, either pH 7.4 and 154mM NaCl, or at lower NaCl concentration (if they are able to estimate it). In the former case, Fig. 4B,D should be replaced by Fig. S4 from Supplementary material and Tab. 1 moved to Supplementary material.
Also, determination of the dissociation constant (p. 9, l. 267ff) is not described very clearly:
(1) What residues were included in fitting? Did the authors exclude data influenced by partial peak overlap (chemical shifts of D99 and R104 at high calcium concentration, cf. Fig. 4B,D and the spectrum in Fig. 2)?
(2) Why the dissociation constant of site 2 was not determined from the chemical shifts of R104 whose peaks are visible at pH 7.4 and 154mM NaCl?
Moreover, I have noticed some formal mistakes and typos:
(3) l. 112 "3D HNCA were..." should read "3D HNCA spectra were..."
(4) l. 154 "and 37 deg C..." should read "at 37 deg C..."
(5) l. 160 there should be a space between number and unit in "405nm".
(6) l. 187 It should be defined what is the orientation of the molecule in panels D and E (one of views shown in panels A-C?)
Author Response
"Please see the attachment."
